# Plasma-induced surface cooling

John A. Tomko [1✉], Michael J. Johnson[2], David R. Boris[3], Tzvetelina B. Petrova[3], Scott G. Walton [3✉] & Patrick E. Hopkins [1,4,5✉]

Plasmas are an indispensable materials engineering tool due to their unique ability to deliver a flux of species and energy to a surface. This energy flux serves to heat the surface out of thermal equilibrium with bulk material, thus enabling local physicochemical processes that can be harnessed for material manipulation. However, to-date, there have been no reports on the direct measurement of the *localized, transient thermal response* of a material surface exposed to a plasma. Here, we use time-resolved optical thermometry in-situ to show that the energy flux from a pulsed plasma serves to both heat and transiently cool the material surface. To identify potential mechanisms for this 'plasma cooling,' we employ time-resolved plasma diagnostics to correlate the photon and charged particle flux with the thermal response of the material. The results indicate photon-stimulated desorption of adsorbates from the surface is the most likely mechanism responsible for this plasma cooling.

[1] Department of Mechanical and Aerospace Engineering, University of Virginia, Charlottesville, VA 22904, USA. [2] Syntek Technologies, Fairfax, VA 22031, USA. [3] Plasma Physics Division, Naval Research Laboratory, Washington, DC 20375, USA. [4] Department of Materials Science and Engineering, University of Virginia, Charlottesville, VA 22904, USA. [5] Department of Physics, University of Virginia, Charlottesville, VA 22904, USA. ✉email: jat6rs@virginia.edu; scott.walton@nrl.navy.mil; phopkins@virginia.edu

Plasmas have long been used for the synthesis[1] and manipulation[2–4] of materials because of their unique ability to deliver both energy and chemically-active species to the surface of materials (Fig. 1)—an attribute that separates them from other approaches to materials processing. Indeed, the energy flux serves to drive the surfaces out of thermal equilibrium with the bulk material, thus enabling local physicochemical processes that can be harnessed to remove (etch) substrate material, deposit different material, or chemically modify the surface. Aside from intentional material modifications, understanding energy delivery at the plasma-surface interface is critical for an array of technologies such as nuclear fusion, where plasma-facing materials must meet complex, yet strict, requirements to avoid degradation from the aforementioned energetic processes[5]. While the benefits or detriments of energy delivery are commonly associated with an increase in temperature, the temperature, is in fact, the net result of the difference between energy delivered to and released from the surface. This can be understood by considering the power balance at the surface[6],

$$P_{\text{in}} - P_{\text{out}} = P_{\text{heat}} \qquad (1)$$

where the power delivered to ($P_{\text{in}}$) and released from ($P_{\text{out}}$) the surface is determined by the flux of energetic particles and radiation arriving at and leaving the surface, along with endothermic and exothermic reactions occurring on the material surface. The difference between $P_{\text{in}}$ and $P_{\text{out}}$ is absorbed by the material ($P_{\text{heat}}$), with a temperature determined by the thermophysical properties of the material. Of course, this power balance does not dictate that the energy delivered to the surface must exceed that released from the surface. The complex array of incident plasma species and chemical reactions at the surface could, in theory, enable local temperatures to both increase or decrease during plasma irradiation. This potential for plasma-induced cooling, or the decrease in the temperature of a surface during plasma irradiation, could provide avenues for structure and device cooling, refrigeration, and temperature-controlled material processing.

Our current understanding of energy delivery from a plasma to a material surface and its response is guided using a variety of ancillary plasma diagnostics[7], steady-state temperature measurements[8,9], models[10,11], and post-treatment ex-situ surface characterization to "re-construct" energy deposition and absorption[6,12,13]. More recently, in-situ materials characterization techniques have been developed that allow for real-time or quasi-real-time analysis[14,15]. While certainly of value, none of these approaches provide a direct measure of the response associated with the flux of species at the surface required to separate the localized and transient energy transport mechanisms from the spatially and temporally averaged net power transfer and temperature rise.

In this work, we experimentally demonstrate the ability of an incident plasma to cool the surface of a material. This cooling is enabled by exposing a surface to a pulsed plasma, which allows the broad range of different energetic processes associated with plasma exposure to be parsed in time. This cooling is then measured through time-resolved, relative temperature changes in the plasma-exposed material with nanosecond resolution.

## Results

In our experiment, we expose a grounded 80-nm gold (Au, 3-nm RMS roughness) film supported by a sapphire substrate to a pulsed, atmospheric plasma jet (Fig. 1) and simultaneously measure the reflectance of a continuous wave laser from the Au surface at the point of contact. The entire system is exposed to the ambient, with room temperature at a constant 20−24°C and relative humidity of 35−45%. A simplified schematic of our experimental configuration is shown in Supplementary Fig. 1.

For the operating conditions in this work, there are negligible laser-plasma interactions, and the reflected beam is not affected by any direct interactions. Rather, we rely on the strong thermoreflectance coefficient of Au at visible wavelengths[16,17] to directly measure the plasma-induced temperature change on the Au surface by means of lock-in detection at the plasma jet repetition frequency, to obtain nanosecond time resolution. While we do not observe any changes in the static reflectivity of the Au surface, plasma effects are further isolated by measuring the differential reflectivity, which is the change in reflectivity of the Au surface relative to the reflectance when no plasma is

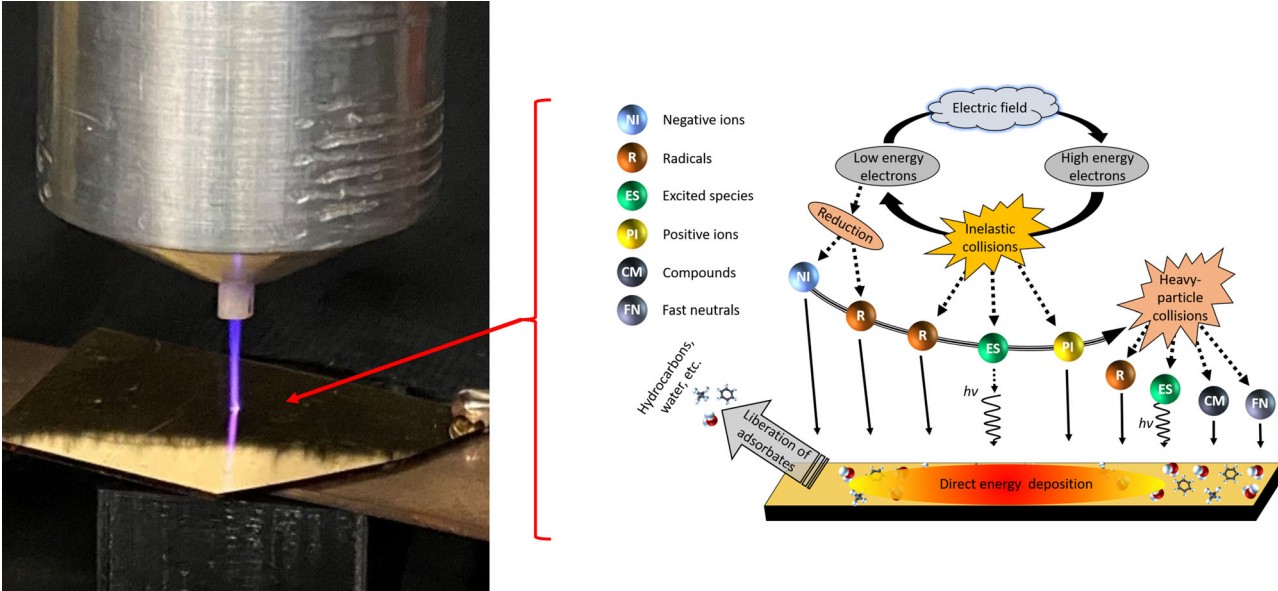

**Fig. 1 Atmospheric jet interactions with metallic surface.** Left. Photograph of the atmospheric plasma jet interacting with a thin Au film on sapphire substrate. Right. Cartoon schematic of the various species and physical processes and resulting species produced within the jet along with their respective interactions with the metal film.

present. We note that the use of lock-in detection at the frequency of which a voltage is applied to the electrode eliminates 'steady-state' effects, such as changes in local gas pressure, that may exist due to the constantly flowing He gas. Additionally, we repeat our thermoreflectance measurements without a voltage applied to the electrode, as well as an applied voltage with no flowing gas (both of which eliminate the production of a plasma jet), and observe neither heating nor cooling—no transient signal is present without the presence of the atmospheric plasma jet.

The use of a thin Au film supported by an insulating substrate satisfies two criteria that are critical to understanding energy transfer at the plasma-surface interface. First, as Au is a noble metal and chemically inert, there are minimal surface reactions (e.g., surface oxidation) that would significantly alter the surface and distort the interpretation of energy deposition mechanisms. Second, because charge transfer and hot-electron effects occur on much faster timescales than investigated in this work, the insulating substrate ensures that charged species and electronically-driven energy transfer from the plasma to the metal surface remain localized to the surface of the Au film. In other words, the electrons and phonons of the Au are thermalized and can be described by their respective thermal distribution functions. This critical aspect ensures that the measured surface temperature is indicative of only the plasma-Au energy transfer and subsequent thermal diffusion rather than ballistic mechanisms that traverse deep into the substrate[18,19].

In considering plasma interactions, there is an incident flux of various species including charged particles, photons, as well as excited and reactive neutrals (Fig. 1), which deliver energy to the plasma-exposed material[6]. In this work, we employ a pulsed, plasma jet produced in helium, which interacts with a gold surface located some distance from the jet (See the supplemental information for more detail). While these sources produce species common to low temperature, non-equilibrium plasmas, there are several unique attributes worth noting that are relevant to the work here. When the high-voltage pulse is applied to the active electrode within the jet body, a high intensity, ionization wave—or "streamer"—is ejected from the jet nozzle and guided by the helium flow through the ambient[20]. As the streamer propagates, it excites, dissociates, and ionizes both the He and air that mixes with it, producing a rich mixture of charged, excited atomic, and neutral (e.g., He, N, O) and molecular (e.g., $N_2$, $O_2$, $H_2O$, OH) species[21]. The streamer velocity is significantly faster than the helium gas flow and quickly terminates at the gold surface located downstream from the jet. The plasma channel left in the wake of the streamer then persists for as long as the voltage is applied to the electrode[21]. The easy to ionize helium column remains remarkably undiluted in the center of the column[22]. As such, species production extends over a fixed volume, but the intensity varies in time. The resulting flux of charged particles - ions of all types and electrons - arriving at the gold film can be measured as a net surface current, which varies considerably in time as shown in Fig. 2a for our experimental system. In these example data, the plasma jet is produced by applying a high voltage (2000 V) for a duration (pulse width) of 5 µs at a frequency of 7.8 kHz. The delay in the rise in current is the time required for the streamer to travel the distance between the powered electrode and the surface. The strong rise and subsequent decay of current at the surface while the voltage is applied is indicative of the streamer colliding with the surface followed by the formation of a weaker plasma column until the voltage is turned off.

Likewise, we can directly measure the temperature change of the material surface (e.g., within the optical skin-depth of the Au film, 15 nm) during plasma irradiation with our thermoreflectance technique since the reflectance of Au is linearly proportional to temperature (e.g., $\Delta R_f = \beta \Delta T$, where $\beta$ is the

thermoreflectance coefficient of Au[23]. An example of our measured thermoreflectance data is shown in Fig. 2b. Note, the thermoreflectance coefficient for this laser wavelength (637 nm) is negative, and thus surface heating corresponds to a decrease in measured reflectance[16,17]. As shown in the Supporting Information (Supplementary Fig. 4), the signal is inverted for probe wavelengths corresponding to photon energies exceeding the interband transition threshold of Au (< 520 nm) and the thermoreflectance coefficient becomes a positive value.

A few salient features can be noted in Fig. 2b. First, at the beginning and end of the plasma pulse, an anomalous decrease in signal magnitude can be observed. This is an artifact in our periodic waveform analyzer and not a true change in optical reflectance due to the plasma pulse; these features are present even when the laser is turned off and the photodetector is blocked. Second, there is a simultaneous, rapid increase of both the measured thermoreflectance and surface current ~2 µs after the rise in voltage. This heating event is followed by a transient decay associated with heat conduction into the substrate, which is governed by the thermal properties of the film and substrate. Interestingly, a peak in reflectivity is observed ~1 µs after the rise in voltage, and prior to the rapid heating event. As the thermoreflectance coefficient of Au at this laser wavelength (637 nm) is negative, this increase suggests a reduction in surface temperature.

As discussed above, the flux of various charged species incident upon the metal film results in a net current, which leads one to invoke Joule's first law where the power dissipated is proportional to the product of the current and resistance (e.g., $P \propto I^2 R$). As with other current sources, this relationship explicitly leads to an increase in temperature, even without consideration of particle interactions (e.g., momentum transfer due to the kinetic energy of incoming ions or the charge transfer associated with neutralization of ions). As the measured thermoreflectance is indicative of a change in the gold's temperature, one should expect that the temporal derivative of this reflectivity trends with the current, since temperature is directly related to particle number density/charge, which is the temporal integrand of current (e.g., $R_f \propto T \propto Q = \int i \cdot dt$). Indeed, as shown in Fig. 2c, the temporal derivative of our thermoreflectance data is in excellent agreement with the measured current. We thus conclude that heating is, as assumed, associated with simple ohmic heating of the metal film. In stark contrast is the strong deviation between current and derivative associated with the observed reduction in temperature prior to heating. This leads to the obvious question: What is the mechanism for this plasma-induced cooling?

## Discussion

There are a number of physical processes arising at the plasma-surface interface that could potentially reduce the temperature of a surface. Thermionic emission, for example, has been theoretically devised as a refrigeration method, with potential efficiencies on par with Carnot cycles[24], though it is nearly impossible to experimentally achieve at or near room temperature, with the exception of limited cases in select material systems such as superlattices[25] and 2-D heterostructures[26]. In the case of an atmospheric plasma jet studied in this work, the gas temperature rises no more than about 20 K for the longest duration plasma pulse (see Supporting Information) and the surface remains at or near room temperature. Thus, the thermally-driven mechanism of electron cooling cannot occur.

However, there are two additional effects that could also lead to the observed cooling phenomenon. The first, which has been recently shown to lead to a temperature decrease in high-repetition laser ablation[27], is material ejection from the surface. It

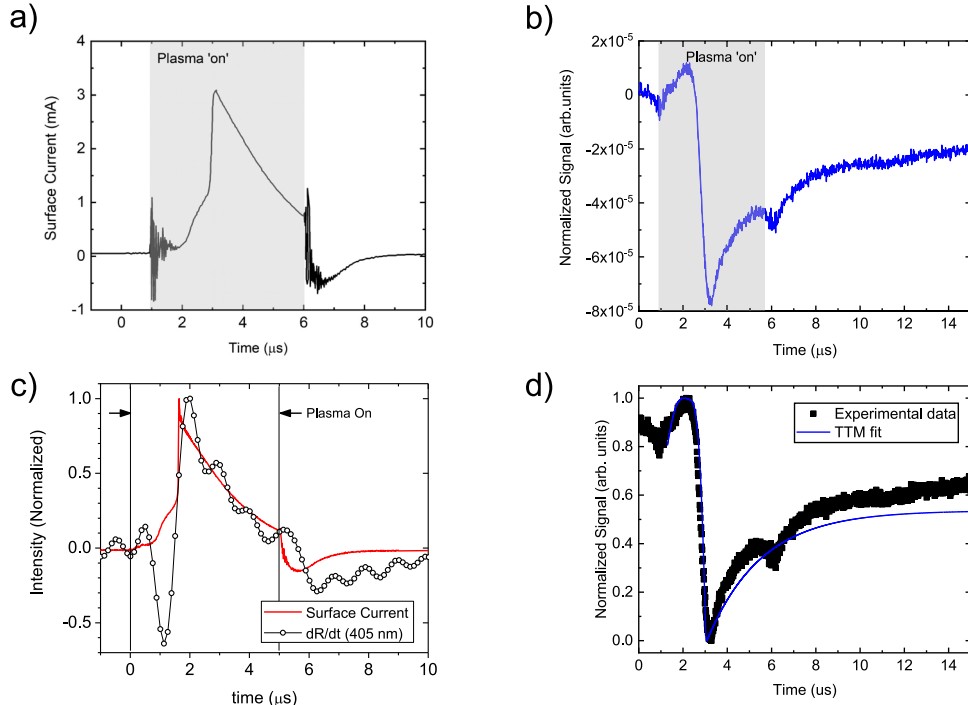

**Fig. 2 Transport measurements of an atmospheric plasma jet. a** The measured surface current of the Au film and **b** the measured thermoreflectance of the Au surface as a function of time for a 5 µs plasma pulse. The laser probe wavelength is 654 nm. **c** Measured surface current (solid red line) overlaid with the temporal derivative of the measured changed in reflectivity due to the plasma pulse (open circles). The laser probe wavelength here is 405 nm. The two are in reasonable agreement, aside from the reduction in surface temperature ~1 µs, indicating Joule heating to be the primary mechanism of energy transfer from the plasma to the Au surface. **d** Measured thermoreflectance data from **b** due to the plasma pulse (black dots) and two-temperature model calculations for the sample system (solid blue line). The observed cooling is attributed to the removal of adsorbed species on the Au surface.

is well-known that plasma jets lead to the modification of a material surface. In fact, when the jet used in this work is operated at higher power (much greater than reported here), erosion of the Au surface is observed. Alternatively, adsorbate desorption could be the underlying mechanism for observed cooling. Despite the inert nature of Au, water will adsorb on Au surfaces even under ultra-high vacuum conditions[28], albeit in a weakly bound physisorbed state (<< 1 eV). In such a case, many species emanating from the plasma jet would deliver enough energy to liberate water molecules. This process is effectively plasma-induced evaporative cooling of the surface.

The second potential mechanism is similar to the Nottingham effect, whereby cooling results from the loss of electrons via field emission[29]. In this work, electrons removed from the surface of the gold during the neutralization of ions as they approach the metal surface[30] or via photoemission may lead to cooling.

To elucidate which of these potential mechanisms is responsible for plasma-induced surface cooling, it is important to consider the magnitude of temperature reduction upon cooling. To gain insight into this, we note that during the observed cooling, the peak differential reflectivity is measured to be $\Delta R_f/R_f \approx 3 \times 10^{-5}$. Based on previous works[16,17], the thermoreflectance coefficients of thin Au films are of a similar order, $\Delta R_f/\Delta T \approx 2-4 \times 10^{-5}$, indicating the observed cooling is on the order of ~1 K.

If we consider the possibility that this temperature reduction is due to the sublimation of the Au film, then only a monolayer-equivalent number of Au atoms would need to be ejected from the film within the plasma-irradiated region to achieve the observed degree of cooling (see Supporting Information for calculations and further discussion on this topic). Similarly, if we consider that the observed cooling is induced by the removal of

adsorbed water, which has a significantly greater heat capacity than that of Au, a similar temperature decrease requires sub-monolayer, or non-uniform distributions, of water to be removed from the surface within the probed volume. This is certainly plausible given our measurements are conducted at standard temperature and pressure[28], where re-absorption occurs on timescales orders of magnitude shorter than the period between plasma pulses (tens of nanoseconds to microseconds for contaminant adsorption, compared to the hundreds of microseconds between each measured plasma pulse). Likewise, a mixture of adsorbed hydrocarbons (e.g., adventitious carbon) and gas would be expected under these conditions. We note that our approximation is likely an overestimation of the average energy-per-particle. Owing to the finite coverage of the adsorbed layers, the phonon population is restricted for measurements at room temperature and is likely restricted to a classical equipartition limit. In addition, adsorbate–adsorbate and adsorbate-substrate interactions (e.g., binding energies) will likely play enough of a role that the energy-per-particle is altered.

We can repeat a similar analysis for the number of electrons that would need to be emitted for a 1 K temperature reduction within the probed volume of the Au; at the timescales measured here, the electron and phonon temperatures can be considered in equilibrium, and thus the energy requirement for a temperature decrease remains dominated by the phononic heat capacity of Au, as used above. Nonetheless, one must consider the energy lost per electron; the average energy of an electron in a metallic system is ~3/5 of the Fermi energy (≈5.5 eV for Au). Thus, for a 1 K temperature decrease at room temperature, $\sim 7 \times 10^8$ electrons would need to be emitted from the Au surface.

To gain further insight into which of these mechanisms, or combination of mechanisms, is driving the observed cooling

process, we perform two-temperature model (TTM) calculations[31] for our experimental geometry. This method allows us to explicitly calculate energy losses/gains to the electron and lattice subsystems. While additional details can be found in the Supporting Information, we use the measured surface current as of the temporal profile for the heating source of the electronic subsystem, and the thermal properties (e.g., thermal conductivity and interfacial thermal resistances) of our 80 nm Au/Al₂O₃ are determined from time-domain thermoreflectance (TDTR) measurements. To simulate the atomic or electronic ejection, we supply a 'cooling' source that removes energy from either the phononic or electronic subsystem. While we ultimately find that either atom or electron ejections can re-produce our experimentally-measured data with high accuracy (see Fig. 2d), these calculations provide important insight into the time-scale of this cooling process. When considering the heating and cooling events together, the best TTM fit to our data requires that the cooling is limited to the first 800 nanoseconds after the voltage is applied. That is to say, the heating and cooling events do not appear to overlap in time.

To properly understand this, we consider the temporal flux of species incident upon the Au surface by comparing the current measurements discussed above with time-resolved photoemission measurements (see measurement details in the Supporting Information). Those results, shown in Fig. 3a, indicate that a fluence of photons impinges upon the Au surface at the timescales corresponding to the cooling and, importantly, they are present prior to the dramatic rise in measured current and heating. These results are characteristic of a surface interacting with a remotely located pulsed plasma jet. While the excited species that relax via photon emission and the charged particles are simultaneously produced near the remotely located electrode when the high voltage is applied, the photons arrive at the surface well before the comparatively low-velocity charged species. The wavelength range of photons emitted from the plasma jet, typical of plasma produced at atmospheric pressure mixtures of air and helium, extends from the IR to the extreme UV, with energies that range from below 1 eV to ≈20 eV (see details in Supporting Information, including Supplementary Figs. 4 and 5). This range of photons incident to the gold surface are certainly sufficient to drive the emission processes discussed above. For gold, the work function is 5.1 eV and the enthalpy of atomization is 3.8 eV/atom (364 kJ mol⁻¹). Likewise, photon-driven desorption of adsorbates[32] has been observed for a wide range of material systems and photon energies. For example, the desorption

of water from Pd(111) has been demonstrated using 6.4 eV (194 nm) and 5.0 eV (248 nm) photons[33]. The liberation of physi-sorbed species such as CO on Ag(111)[34] and H₂CO or CH₂CO on Ag(111)[35] only requires 1.1 eV (1098 nm) photons. While these species desorb as neutrals, oxygen can also be liberated from aluminum in the form of negative ions by photons in the range of ≈8–10 eV[36].

The possibility of a photon-driven cooling mechanism is further supported by spatially-resolved thermoreflectance measurement results shown in Fig. 3b. For these data, we raster the plasma jet with respect to the position of the laser probe, allowing us to extract a spatio-temporal temperature profile of the Au surface. In agreement with our previous work[14], we observe a heating profile of ≈0.5 mm; this is ≈1/3 of the tube diameter from which the jet emanates. More importantly, in this work, we observe the cooling of the Au surface in regions extending beyond the heated width. This can be understood by recognizing the plasma is 'guided' by the helium flow leaving the tube, such that the majority of the plasma-produced particles arrive at the surface over an area dictated by the diameter of the gas flow/surface intersection. Conversely, the photons formed via spontaneous emission from excited species formed along the plasma channel have both random phase and direction, thus potentially interacting with a much larger surface area. The exception to this will be sub-200 nm photons, which are readily absorbed in the air outside the helium channel. Still, it is reasonable to expect a concentrated region of charged particle interactions toward the center and a more diffuse region of photon interactions extending beyond the region dominated by charged particles.

While a photon-driven process is reasonably supported by the results, the liberation of substrate material and surface adsorbates, as well as electronic emission, can all be supported by our measurements, two-temperature analysis, and prior TDTR measurements. Discerning the potential contribution of these processes requires additional considerations.

We start by noting the results of Fig. 3a clearly show the photons arrive at the surface during the entire voltage pulse, as well as after the voltage is extinguished. That is, photons are present both before and after the charged particle flux that is associated with heating. If the cooling were caused by photon-driven electron emission or ejection of gold atoms, one should expect cooling to occur during both times that photons are the dominant species since there is ample material mass and essentially an infinite reservoir of electrons

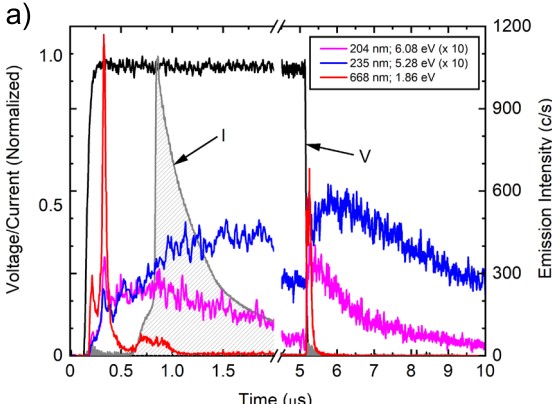
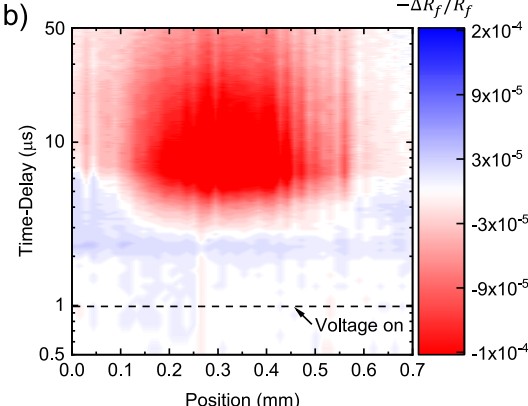

**Fig. 3 Time-resolved emission and in-plane thermal transport dynamics. a** Time-resolved emission measurement of the He jet interacting with the gold surface along with the voltage applied to the electrode and the measured surface current. The emission lines shown are from He* at 668 nm and NO* at 204 nm and 235 nm, illustrating that a range of energetic photons arrives at the surface both before and after the charged particle flux. **b** Spatially-resolved thermoreflectance measurements of an Au surface. A width of 0.5 mm is heated from the flux of charged particles, while the observed photon-induced cooling extends to a much larger region. Blue corresponds to the observed cooling of the sample surface (increase in the modulated reflectance signal, ΔR_f), while red denotes an increase in surface temperature (decrease in ΔR_f).

to be ejected. And yet, cooling is only observed before heating. On the other hand, photo-desorption of adsorbed species would only occur during the initial flux of photons since the flux of all species—particularly ions—will remove adsorbates from the surface prior to the end of the applied voltage pulse. Thus, photon-stimulated desorption of adsorbates is the most likely mechanism responsible for plasma cooling.

To further reinforce this plasma-induced evaporative cooling of a metal surface, we consider the relative heating and cooling contributions with varying plasma jet parameters. For example, as the applied voltage of the electrodes is increased, we observe a corresponding increase in the measured surface current and the associated peak temperature of our thermoreflectance measurements (see Supporting Information and Supplementary Fig. 6); these observations can be attributed to an increase in the flux of charged species at the surface associated with a higher density plasma formed with increasing voltage. However, the minimum temperature achieved during cooling is negligibly affected by these changes, at least within the regimes considered in this work. Yet, like the flux of charged particles, the flux of photons responsible for either desorption or electronic emission will be increasing with voltage as well. In other words, the cooling magnitude has saturated and the additional flux does not play a role. As noted previously, neither gold atoms nor electrons are in limited supply, and so atom- or electron-mediated cooling mechanisms should not plateau with increasing photon flux. This further reinforces our position that neither Au atom nor electron ejection is responsible for the observed plasma-induced cooling. A similar result is observed when varying voltage pulse widths, where the peak temperature varies significantly with changes in charged particle flux. The net cooling magnitude, however, remains relatively constant above some critical width ($\approx 2\,\mu s$ in this work). Interestingly, below this critical pulse width, this cooling can occur without any significant plasma-induced heating of the Au surface. Here, a sufficient number of high-energy photons are produced at plasma ignition to drive desorption, but the voltage is extinguished before the streamer and the associated flux of charged particles can reach the surface with sufficient intensity. Just above this temporal width, a spike in both the heating and cooling profile is observed due to an abundance of both charged particles and photons. While these observations clearly reinforce the notion of photon-driven, desorption-induced cooling of the Au surface, the complexities of experiments with plasma jets at ambient conditions limit the number of controls required to identify cooling mechanisms with further certainty. It will be interesting to investigate these phenomena with different materials and/or gas compositions to see how cooling varies. Still, the results indicate potential regimes for which the relative heating and cooling profiles of the gold surface can be controlled by manipulating various plasma parameters. In fact, our previous work[14] identified conditions where net cooling was a possibility, but in the absence of an explanation, it remained a curiosity. The results shown here demonstrate when cooling occurs and how it can be controlled.

In summary, we have provided a nanosecond-resolved measurement of energy transduction during pulsed plasma interactions with a surface using a time-dependent thermoreflectance method. The heating of the metal Au surface through a flux of charged species is the predominant mechanism of energy transfer, which dissipates primarily through conduction into the material; this process is in excellent agreement with our TTM calculations. Furthermore, we find that the plasma jet induces transient cooling of the Au surface prior to heating. This cooling is correlated to a flux of photons that precedes the charged particle flux, which we argue, liberates surface adsorbates from the Au film, thus temporarily cooling the metal surface; this process is typically overwhelmed by the heating of material surfaces during steady-state operation and

thus previously undetectable. While more work is needed to fully understand the underlying mechanisms, the results open the door to a previously unreported means of surface cooling and provide insight into the plasma-material interactions that drive material modification and processing.

## Methods

**Atmospheric plasma jet**. A schematic of our atmospheric plasma jet and the experimental setup is shown in Supplementary Fig. 1a. The details of this experimental configuration have been discussed in detail in our previous work[14]. In summary, the plasma jet consists of a hollow needle electrode centered within a ceramic tube; this tube is enclosed by a grounded, metal casing. Helium flows through the needle electrode at a constant rate (1500 sccm; measured with a mass flow controller). To drive plasma production, high-voltage pulses ($1.5-2\,kV$) are applied to the needle for a set duration ($2-5\,\mu s$) and frequency. For the data in this work, the operating voltage, pulse width, and frequency were $2\,kV$, $5\,\mu s$, and 7.79 kHz, unless noted otherwise. Our experiments are performed in laboratory air with a relative humidity of $35-45\%$. The measurement setup is partially enclosed to minimize external flow from laboratory air conditioners and filtration systems that can perturb the plasma jet during laser spectroscopy measurements. The substrate in this work is an 80 nm Au film on a crystalline $Al_2O_3$ substrate; the metal film is located 1 cm from the exit of the jet and connected to the ground. Current transformers and high-voltage probes are used to measure the current to the ground and power to the driven electrode.

**Transient laser reflectivity measurements**. To measure the temperature of the Au surface during plasma interactions, we perform transient thermoreflectance measurements with a laser probe, as illustrated in Supplementary Fig. 4. For this work, we focus an incident laser, with a wavelength of either 405 nm or 637 nm, to the Au surface at an incident angle of ≈45 degrees. The reflected beam is collimated and re-focused into a silicon photodiode. The photodiode output is then sent to a pre-amplifier; this output is then measured with a lock-in amplifier (Zurich Instruments UHFLI). We use the applied voltage for the plasma production as the reference signal for the lock-in amplifier. However, rather than measuring the magnitude/phase of the reflectance, we use periodic waveform analysis (comparable to moving boxcar averaging) to obtain the transient differential reflectance that is induced by the plasma jet.

**Photoemission measurements**. To understand the relative flux of species at the surface, optical emission spectroscopy was employed to measure both time-averaged broadband emission as well as time-resolved emission intensity of select lines. The geometry for these measurements is shown in Supplementary Fig. 4. A NIR spectrometer (Ocean Optics HR2000+) was used to measure the time-averaged emission spectrum from the jet using the geometry shown in Supplementary Fig. 4a. Those results are shown in Supplementary Fig. 5 and indicate a wide range of photons - extending from UV to IR—are produced by the plasma jet. VUV and EUV photons (<200 nm) have also been observed in He jets, but are not observable in our experiments due to bandwidth limitations and strong absorption in air. The spectrum also indicates that, despite being produced in a helium flow, a wide variety of air-related products are produced in the plasma due to the downstream mixing of helium and air. While useful for species identification, the spectrum lacks any spatial or temporal information. To best understand the reflectivity results, the time-resolved flux of photons at the surface must be measured. To do this, we employed the geometry shown in Supplementary Fig. 4b, which provides a line-of-sight along the axis of the jet and thus the ability to measure emission incident to the surface from ignition through the afterglow. Select emission lines were monitored using a scanning monochromator (2035 McPherson) with a photomultiplier tube (McPherson model number 654) connected to a multichannel scaler (Stanford Research Systems Model SR430 multichannel scaler) to accumulate the data. Those results are shown in Fig. 3 of the main article and compared to the time-dependent voltage and current measurements.

## Data availability

The data that support the plots within this paper and other findings of this study are available from the corresponding authors.

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

## Acknowledgements

This manuscript is based upon work supported in part by the Office of Naval Research under award No. N00014-20-1-2686 and by the Air Force Office of Scientific Research under award no. FA9550-18-1-0352. This work was also partially supported by the Naval Research Laboratory base program.

## Author contributions

J.A.T. and S.G.W. performed the time-resolved optical thermometry measurements. M.J.J. and D.B.R performed plasma characterization measurements and the corresponding analysis. T.B.P. provided theoretical support for plasma characterizations. J.A.T. performed the thermal model calculations. S.G.W. and P.E.H. conceived the experiments and supervised the project. All authors discussed the results and assisted in writing the manuscript.

## Competing interests

The authors declare no competing interests.
