## [Peer Review File · Nature Communications]

Plasma-induced surface coolingREVIEWER COMMENTS

Reviewer #1 (Remarks to the Author):

The paper presents that plasma can be used for surface cooling.

The title looks attractive. However, there are too many scientifically unclear issues in the submitted version, and the paper is not ready for a scientific review.

There are attempts to measure doses from a plasma using for example ESR, which is missing in state of art.

A plasma is generated by a cold plasma torch. A pulsed square wave voltage is applied. The current should usually abruptly increase until the voltage becomes zero. It is not clear why the current suddenly drops down while the voltage is still high.

The experimental method is also quite unclear. For example, is the gold connected to the ground, or is it floating? How the sapphire substrate is held? Is it cooled during experiment? What are room temperature, substrate temperature, and gas temperature? The gas temperature must be lowered when the gas is fed to the atmospheric air due to the slight pressure drop.

The authors argue that the major cause of the temperature increase is the joule heating of the gold surface. Why the heat irradiation from the plasma can be neglected? The temperatures of the plasma and the surrounding gas should be measured and reported.

The authors discuss that the main cause of the cooling can be evaporation of water etc. on the gold surface. If so, it can always occur with or without a plasma. In addition, when water is adsorbed during no plasma, there must be a temperature increase. How this phenomenon can be integrated with the argument presented in the paper?

These are just examples the reviewer immediately questions. After they are clearly presented, this paper can be more carefully reviewed.

Reviewer #2 (Remarks to the Author):

Referee's comment:

This paper titled "Plasma Cooling" describes the cooling phenomenon of the gold surface when the He plasma jet impinges on it. The referee agrees that a certain degree of novelty for this interesting

phenomenon. As the authors described in the manuscript, this cooling phenomenon is very interesting and it has potentials for the future applications. The authors insisted that the cooling mechanism is photo-desorption of adsorbed species. The referee agrees the possibility that incident photons to the surface may drive the desorption process. However, the referee feels that the discussion in this paper is not enough to convince the readers of the light-driven desorption process. Thus, the referee will not be able to recommend this manuscript for the publication in Nature Communications.

The referee's questions / comments are listed below;

1) In this paper, the cooling phenomenon is caused by the photo-desorption of adsorbed materials by the incident light, and the incident of the plasma jet to the surface is only treated as an electron current source heating the surface. If so, does the cooling phenomenon shown in this manuscript occur when only the light is irradiated to the surface without the impinge of the plasma jet? Is such an additional experiment possible?

2) If the desorption of adsorbed materials by the light cools the surface of gold, the total amount of the removed energy is supposed to depend on the amount of adsorbed material within the skin depth of the light. If a fresh surface without adsorption is used for the experiment, does the cooling phenomenon disappear? Is such an additional experiment possible? The referee feel that the sufficiently strong evidence or discussion is not provided for the authors' claims about the cooling mechanism driven by the photo-desorption.

3) In the figure 2(d), the calculation by TTM agrees very well with the experiment. This seems to support the validity of the authors' claim quantitatively. However, the parameters used for the calculation by TTM are not shown at all in the manuscript and supporting document. The referee would ask the authors to describe the detail of the values used in TTM calculation. In the current manuscript, it is very difficult for the readers to quantitatively trace the authors' analysis of the cooling mechanism.

4) In the supporting document, the authors describe that [0.37nJ must be removed from the probed volume to cause a local 1K temperature drop]. This is correct if the area of interest is thermally isolated from the surrounding condition, but the thermal conduction occurs in the actual situation. Even if the energy is removed locally, the surface temperature may not drop if heat is supplied by heat conduction from the surroundings. A more quantitative and detailed explanation of temperature drop is needed.

5) In Figures 2(a)-(d), the time origin of the horizontal axis should be the same.

6) In the line 223, page 10 of the manuscript, authors described “(see details in Supporting Information, including Figs. S3-5).” Is referring to Fig. S3 correct here?

7) In the reference to supporting document, the page number is lack for the reference 5.

Reviewer #3 (Remarks to the Author):

The authors describe and demonstrate a surface cooling effect induced by an atmospheric plasma jet impinging on a Au surface. While the effect is small (~ 1 K), is of interest from a fundamental physics and surface chemistry perspective. The paper is well-written, clearly structured, concise yet rather complete. Overall, I find this a very nice paper.

The cooling effect is discussed to be the result of two possible mechanisms: surface species evaporation and the Nottingham effect. The authors present experimental evidence, supplemented by two-temperature model calculations, for a photon-driven cooling due to adsorbate desorption, and at the same time excluding photon-driven electron emission and Au-atom ejection.

To the best of my understanding, all experiments were carefully designed and executed. The conclusions are in line with the presented data and discussion.

Still, the paper could potentially benefit from considering the following remarks.

1. In the introduction, it would be useful to clearly define what is considered as “the surface”, or the “surface/plasma interface”. While the Au film is only 80 nm thick, surely this entire 80 nm cannot be considered as “the surface”. I imagine that a (time-dependent) temperature gradient must exist across the film.

2. Moreover, Au is used because of its noble character. However, at the nanoscale, Au is reactive, and in fact used as a catalyst. This reactivity is larger as the surface density of defects such as steps and kinks increase. Atomically flat surfaces are less prone to reaction. Thus, it would be useful to have a metric on the surface flatness.

3. Very important for understanding the plasma/interface interaction is the knowledge of which species and densities thereof actually reach the surface. There is however no so much information provided on this; is the air (molecular species) excited or ionized? Is there any information on the electron densities and the electron energy distribution near the surface? Same question for the IEDF. I understand that the streamer reaches the Au film, but to me it is not so clear if only electrons and He⁺ ions (and other He-based excited species) reach the surface, or also other species.

4. As a follow-up on the previous question, molecular plasmas behave very differently because a significant amount of the plasma power is used to excite species into vibrational and rotational levels. As most plasmas for applications use molecular gases, I wonder if the conclusions reached here would still be maintained in such case?

5. In the beginning of the Discussion, the authors state “Although the notion of cooling resulting from an incident energy flux is seemingly counter-intuitive, ...” However, I would like to remark that even for a system in thermal equilibrium, microscopic balance and the 2nd law of thermodynamics dictate that the flux of species desorbing from the surface have a non-Maxwellian energy distribution, with a mean energy below that of the gas phase species, in case the impinging species have an energy-dependent sticking coefficient below 1 – aka, translational cooling in desorption. Of course, this does not mean that the surface is being cooled. I therefore wonder if the wording is or could be confusing to readers.

Reviewer #1 (Remarks to the Author):

The paper presents that plasma can be used for surface cooling. The title looks attractive. However, there are too many scientifically unclear issues in the submitted version, and the paper is not ready for a scientific review.

There are attempts to measure doses from a plasma using for example ESR, which is missing in state of art.

The reviewer brings up a good point. However, it is our opinion that a full analysis of this type is not required for the work. For this effort, a power balance approach following the seminal work of Holger Kersten and his colleagues is appropriate to account for energy transfer kinetics. Even so, a full analysis of this type will not add to the results presented in this work. The main point of the paper is the observed cooling, which precedes the expected results, which are plasma-induced heating.

A plasma is generated by a cold plasma torch. A pulsed square wave voltage is applied. The current should usually abruptly increase until the voltage becomes zero. It is not clear why the current suddenly drops down while the voltage is still high.

The voltage presented in the manuscript is the voltage applied to the electrode. The current is measured on the grounded surface and *not* the driven electrode. For a pulsed plasma jet with a duration of several microseconds, the voltage and current profiles presented are in line with those found in the literature. We have fixed some of the language in the paper to more clearly identify where the voltage and current are measured, including the experimental schematic in Fig. S1.

The experimental method is also quite unclear. For example, is the gold connected to the ground, or is it floating? How the sapphire substrate is held? Is it cooled during experiment? What are room temperature, substrate temperature, and gas temperature?

We appreciate the reviewer's comments. While some of these details were in the Supporting Information, we have moved it to the main document. In addition, we have added measured rotational and vibrational temperature information for several operating conditions in the Supporting Information. As the reviewer is aware, the rotational temperature can be used as a proxy for gas temperature and so we have noted in the main manuscript that the temperature does not rise significantly.

The revised text in the main manuscript now reads as:

“Specifically, we expose a grounded 80 nm gold (Au) film supported by a sapphire substrate to a pulsed, atmospheric plasma jet and simultaneously measure the reflectance of a continuous wave laser from the Au surface at the point of contact. The

entire system is exposed to the ambient, with room temperature at a constant ~20-24 C and a relative humidity of ~35-45%. A simplified schematic of our experimental configuration is shown in Fig. S1.”

The revised text in the Supporting Information now reads as:

“...Our experiments are performed in laboratory air with a room temperature of 20 – 24°C and a relative humidity of 35% – 45%. The measurement setup is partially enclosed to minimize external flow from laboratory air conditioners and filtration systems that can perturb the plasma jet during laser spectroscopy measurements. The workpiece in this work is an 80 nm Au film on a crystalline Al₂O₃ substrate. The entire workpiece is suspended in front of the plasma jet by an insulating holder such that the metal film is located 1 cm from the exit of the jet and connected to ground. There was no attempt to cool the workpiece. Current transformers and high voltage probes are used to measure the substrate current (through the lead to ground) and power to the driven electrode.”

The gas temperature must be lowered when the gas is fed to the atmospheric air due to the slight pressure drop.

This is a critical aspect of our measurement technique. By using lock-in thermorefectance, our measurement detects only the *transient* change in temperature induced by a given frequency. In this work, the detected frequency is given by the frequency of which a voltage is applied to the electrode and a plasma is generated. In other words, our lock-in method measures *only* the change in temperature that repeatedly occurs at ~10 kHz. However, the gas flow is *constant* as a function of time, and does not have a time varying component; thus, the gas is ‘undetected’ by our measurement technique and thus cannot be responsible for the observed cooling. To further support this aspect, we repeat our measurements with no plasma generation (i.e., no applied voltage to the electrode), while still flowing the gas, and see no effect – there is neither heating nor cooling of the sample surface in this control experiment.

To increase clarity to potential readers, we have added these details to our main manuscript. The revised text can be found in the description of our experimental design, and now reads as:

“Rather, we rely on the strong thermorefectance coefficient of Au at visible wavelengths to directly measure the plasma-induced temperature change on the Au surface by means of lock-in detection at the plasma jet repetition frequency, to obtain nanosecond time resolution. While we do not observe any changes in the static reflectivity of the Au surface, plasma effects are further isolated by measuring the differential reflectivity, which is the change in reflectivity of the Au surface relative to the reflectance when no plasma is present. We note that the use of lock-in detection at the frequency of which a voltage is applied to the electrode eliminates ‘steady-state’ effects, such as changes in local gas pressure, that may exist due to the constantly flowing He gas. Additionally, we repeat our thermorefectance measurements without a voltage applied to the electrode, as well as an applied voltage with no flowing gas (both of which eliminate production of a plasma

jet), and observe neither heating nor cooling - no transient signal is present without the presence of the atmospheric plasma jet.”

The authors argue that the major cause of the temperature increase is the joule heating of the gold surface. Why the heat irradiation from the plasma can be neglected? The temperatures of the plasma and the surrounding gas should be measured and reported.

This is a great question. We did not mean to suggest irradiation or other mechanisms that can heat a surface were ignored or neglected. Rather, we see a strong correlation between the heating profile and the surface current. That is to say, given the time-dependent heating profile overlays the derivative of the reflectivity so well (with the exception of the cooling part), we can attribute *majority* of heating to be induced by the flux of charged particles to the metal surface. Any neutral gas dynamics associated with heating will have very different time profiles, and do not appear to play a large role in the heating dynamics *at the timescales investigated in this work*.

To increase clarity to potential readers, we have added additional information on the gas and plasma temperature to both the main manuscript as well as the Supporting Information. Specific details on the plasma temperature, added to the SI, now read as:

“While it is difficult to make a non-intrusive measurement of the gas temperature within an atmospheric pressure plasma, the rotational temperature of N₂ can be used as an approximation. To measure the rotational temperature, the emission from the second positive system was measured over the entire voltage pulse at the plasma jet-substrate interface and is shown in Fig. S6. This spectrum was compared to a simulated spectrum generated that followed a Boltzmann distribution generated using the software package ‘MassiveOES’. The rotational and vibrational temperature that produced the simulated spectrum closest to the measured spectrum was determined and is shown in Table S1. Regardless of the length of the voltage pulse, the rotational temperature never gets significantly higher than room temperature, as it reaches ~314 K over the longest tested pulse width. This approximation is based on a time-averaged measurement of the N₂ rotational lines during the entire pulse duration. Accordingly, this estimate should be considered as average gas temperature during the time the plasma is active.

Figure S6. The measured and simulated emission spectra of the 337 nm line of the second positive system in N_2 [$N_2(C^3\Pi_u) \rightarrow N_2(B^3\Pi_g)$].

pulse width (μ s)	rotational temperature (K)	vibrational temperature (K)
3	301.55	1040.1
5	311.85	1004.9
7	309.04	1018.1
10	314.36	1120.3

Table S1. The approximate rotational and vibrational temperature taken from the 337 nm line in the second positive system for a given duration of the applied voltage.”

The authors discuss that the main cause of the cooling can be evaporation of water etc. on the gold surface. If so, it can always occur with or without a plasma. In addition, when water is adsorbed during no plasma, there must be a temperature increase. How this phenomenon can be integrated with the argument presented in the paper?

This is a critical aspect of our measurement technique, and similar to the reasons we do not see the net reduction in temperature induced by the constantly flowing gas. By using lock-in thermorefectance, our measurement detects only the *transient* change in temperature induced by a given frequency. In this work, the detected frequency is given by the frequency of which a voltage is applied to the electrode and a plasma is generated. In other words, our lock-in method measures *only* the change in temperature that repeatedly occurs at ~ 10 kHz. However, the gas flow is *constant* as a function of time, and does not have a time varying component; thus, the non-plasma-induced changes in water adsorption are ‘undetected’ by our measurement technique and thus cannot be responsible for the observed cooling.

To increase clarity to potential readers, we have added these details to our main manuscript. The revised text can be found in the description of our experimental design, and now reads as:

“Rather, we rely on the strong thermorefectance coefficient of Au at visible wavelengths to directly measure the plasma-induced temperature change on the Au surface by means of lock-in detection at the plasma jet repetition frequency, to obtain nanosecond time resolution. While we do not observe any changes in the static reflectivity of the Au surface, plasma effects are further isolated by measuring the differential reflectivity, which is the change in reflectivity of the Au surface relative to the reflectance when no plasma is present. We note that the use of lock-in detection at the frequency of which a voltage is applied to the electrode eliminates ‘steady-state’ effects, such as changes in local gas pressure, that may exist due to the constantly flowing He gas. Additionally, we repeat our thermorefectance measurements without a voltage applied to the electrode, as well as an applied voltage with no flowing gas (both of which eliminate production of a plasma jet), and observe neither heating nor cooling - no transient signal is present without the presence of the atmospheric plasma jet.”

These are just examples the reviewer immediately questions. After they are clearly presented, this paper can be more carefully reviewed.

We appreciate the comments and suggestions, and, as a result of these, believe our revised manuscript is greatly improved from our initial submission.

Reviewer #2 (Remarks to the Author):

This paper titled “Plasma Cooling“ describes the cooling phenomenon of the gold surface when the He plasma jet impinges on it. The referee agrees that a certain degree of novelty for this interesting phenomenon. As the authors described in the manuscript, this cooling phenomenon is very interesting and it has potentials for the future applications. The authors insisted that the cooling mechanism is photo-desorption of adsorbed species. The referee agrees the possibility that incident photons to the surface may drive the desorption process. However, the referee feels that the discussion in this paper is not enough to convince the readers of the light-driven desorption process. Thus, the referee will not be able to recommend this manuscript for the publication in Nature Communications.

The referee's questions / comments are listed below;

1) In this paper, the cooling phenomenon is caused by the photo-desorption of adsorbed materials by the incident light, and the incident of the plasma jet to the

surface is only treated as an electron current source heating the surface. If so, does the cooling phenomenon shown in this manuscript occur when only the light is irradiated to the surface without the impinge of the plasma jet? Is such an additional experiment possible?

This is a great question. We repeated this experiment with visible irradiation rather than a plasma jet, and do not see cooling. However, this is consistent with a large number of pump-probe studies (including our own): the majority of visible photons do not interact with surface adsorbates, but rather couple directly to electrons within the metal film. This results in only a temperature increase, with no observed cooling. While we think this study repeated at greater photon energies would be an excellent addition to this work, it is *incredibly* difficult to generate deep-UV (or greater photon energy) wavelengths with consistently high output frequency or short temporal widths. Specifically, we have purchased a number of lasers to attempt this experiment, but the inconsistency in output frequency makes lock-in detection unfeasible (e.g., a 250 nm laser source with a repetition rate of 10 Hz, but drifts every few seconds to either 9 or 11 Hz causes too large of error for any meaningful data acquisition). Similarly, the 'pulsed' generation of light would be critical, as, once adsorbates are removed, the remaining photons would directly excite electrons in the metal and lead to the aforementioned heating.

2) If the desorption of adsorbed materials by the light cools the surface of gold, the total amount of the removed energy is supposed to depend on the amount of adsorbed material within the skin depth of the light. If a fresh surface without adsorption is used for the experiment, does the cooling phenomenon disappear? Is such an additional experiment possible? The referee feel that the sufficiently strong evidence or discussion is not provided for the authors' claims about the cooling mechanism driven by the photo-desorption.

This is a great suggestion by the reviewer. The correct way to do this would be in an ultra-high vacuum chamber with a controlled dose impingement upon the surface. However, this is not achievable with an atmospheric plasma jet; specifically, a jet cannot be formed in low pressures.

While we agree that our results are simply not one-hundred percent conclusive with regards to the mechanism of photo-desorption, we strongly believe the strength of this work lies in this being the first-known observation of plasma-induced cooling; a regime that was not known to exist prior. With that in mind, through the array of measurements performed that are accessible with our current experimental geometry, our results *suggest* photo-desorption to be the primary mechanism. To this end, we have edited the language in our revised manuscript to soften the 'definitive' phrasing in our initial manuscript, and re-focus the discussion on the *observation* of plasma-induced cooling, that we believe to be the merit of this work.

3) In the figure 2(d), the calculation by TTM agrees very well with the experiment.

This seems to support the validity of the authors' claim quantitatively. However, the parameters used for the calculation by TTM are not shown at all in the manuscript and supporting document. The referee would ask the authors to describe the detail of the values used in TTM calculation. In the current manuscript, it is very difficult for the readers to quantitatively trace the authors' analysis of the cooling mechanism.

We appreciate the reviewer's comment and suggestion. We have refined the TTM discussion in the Supporting Information to increase clarity to potential readers, including a Table of all relevant thermophysical parameters.

4) In the supporting document, the authors describe that [0.37nJ must be removed from the probed volume to cause a local 1K temperature drop]. This is correct if the area of interest is thermally isolated from the surrounding condition, but the thermal conduction occurs in the actual situation. Even if the energy is removed locally, the surface temperature may not drop if heat is supplied by heat conduction from the surroundings. A more quantitative and detailed explanation of temperature drop is needed.

The reviewer is correct that heat conduction occurs and affects the nominal value. This calculation assumes an *instantaneous* removal of energy via some cooling mechanism, with the intent to provide the order of magnitude of energy and the volume of associated material removal that would be necessary. Prior to the observed cooling, there is no observed heating of the material from the plasma source. Thus, the only 'source' of thermal energy is from the uncooled region. In other words, as suggested by the reviewer, if cooling occurs over an 800 ns window, heat conduction from the now-hotter surrounding substrate will lead to an under-estimation of the energy required for a 1 K reduction in the peak temperature. In other words, we would anticipate *greater* volumes of either material or adsorbates to be removed.

Indeed, our best-fit results for the one-dimensional TTM that provides good agreement with the experimental data suggests that a significantly greater amount of energy is removed from the system (recall: we include an additional source term in which energy is removed, rather than added, to the system in our model). Specifically, the energy is on the order of ~25 nJ of energy removed over the course of 800 ns. If we were to recalculate the volume of Au necessary to be removed, we would find that significant damage to the Au film would occur if material ejection were the cause for cooling, which would further support the posit of adsorbates being the primary mechanism for cooling. However, as noted in our revised manuscript, these calculations provide simply *estimates* for the energy removed – due to the geometry of the plasma source relative to the laser probe, as well as the time-scales considered (hundreds of nanoseconds to microseconds), the full 3-dimensional nature of the system likely distorts the absolute values. However, we strongly believe it provides insight to the relative range of necessary material removal for such cooling to occur (~monolayer to, at the upper bound, tens of nm).

In our revised text, we address this critical aspect of our assumptions and models used in this work. In the Supporting Information, this section now reads as:

“To understand how cooling via either an electronic or atomic channel affects the temporal dynamics, we introduce a second ‘source’ term to either Eq. 1 or Eq. 2, respectively. To mimic electronic ejection, we simply implement a Gaussian cooling event that reduces the surface temperature of the electronic subsystem, with a peak temperature decay of 1 K. This is mimicked for the case of adsorbate removal, where the cooling term is inserted to Eq. 2. In both cases, the best-fit occurs with a time-scale of cooling of ~800 ns. However, we note that due to the time scales investigated in this work, our thermoreflectance measurement is insensitive to whether the cooling occurs through electronic or atomic processes; a temporal resolution on the order of 100s of femtosecond to a picosecond would be necessary to directly interrogate and separate these processes with laser-based spectroscopy techniques.

In the case of material removal for evaporative cooling, we can directly calculate the theoretical energy loss (i.e., change in temperature) associated with the process of atomic desorption from the target surface based on the specific heat of species within our probed measurement volume. Based on the spot size (~55 μm) and skin depth (~15 nm) of our laser in Au, the probed volume is ~150 μm^3 . Pairing this volume with the volumetric heat capacity of Au, ~2.49 $\text{MJ m}^{-3} \text{K}^{-1}$, a local 1 K temperature reduction would require 0.37 nJ to be removed from the probed volume. At room temperature, a reasonable approximation to the average thermal energy of each particle in a solid is simply $3k_{\text{B}}T$, or ~ 1.24×10^{-20} J. With these values in mind, it would require 3×10^{10} atoms to be removed from the probed volume; this number of atoms is equivalent to approximately a single monolayer of material removed homogeneously across the area of the laser probe. If we consider that the cooling is induced by the removal of adsorbed water, which has a heat capacity of ~4.2 $\text{MJ m}^{-3} \text{K}^{-1}$, a similar temperature decrease requires sub-monolayer thicknesses, or non-uniform distributions, of water to be removed from the surface. When considering a similar calculation for electron ejection, the energy removed from the probe volume would remain 0.37 nJ.

We note that the above estimations assume an instantaneous removal of energy at the peak temperature reduction for an estimate on the order of magnitude required for our observations. For cooling over the course of 800 ns, heat conduction from the now-hotter surrounding substrate will lead to an under-estimation of the total energy required for a 1 K reduction in the peak temperature. In other words, we would anticipate larger volumes of material or adsorbates to be removed. Indeed, our best-fit results for one-dimensional TTM calculations, which provide good agreement with our experimental data, suggest that a significantly larger amount of energy is removed from the system; over the course of 800 ns, approximately 25 nJ of energy would have to be removed. Returning to the above calculations of material/adsorbate removal, this energy would require very large volumes of Au removal, further supporting the posit of adsorbate evaporation from the Au surface. However, we note that this calculation is simply an estimate and does not account for the 3-dimensional nature of heat conduction and plasma interactions (recall: the plasma source is significantly larger than the laser probe).”

5) In Figures 2(a)-(d), the time origin of the horizontal axis should be the same.

We have updated the time origin for the surface current plots to be the same, as well as the differential reflectivity plots.

6) In the line 223, page 10 of the manuscript, authors described “(see details in Supporting Information, including Figs. S3-5).” Is referring to Fig. S3 correct here?

The figure label was incorrect, and has been revised to correctly read as “...including Figs. S4 and S5.”

7) In the reference to supporting document, the page number is lack for the reference 5.

Thank you for noting this; the page number has been added to the reference.

Reviewer #3 (Remarks to the Author):

The authors describe and demonstrate a surface cooling effect induced by an atmospheric plasma jet impinging on a Au surface. While the effect is small (~1 K), is of interest from a fundamental physics and surface chemistry perspective. The paper is well-written, clearly structured, concise yet rather complete. Overall, I find this a very nice paper.

The cooling effect is discussed to be the result of two possible mechanisms: surface species evaporation and the Nottingham effect. The authors present experimental evidence, supplemented by two-temperature model calculations, for a photon-driven cooling due to adsorbate desorption, and at the same time excluding photon-driven electron emission and Au-atom ejection.

To the best of my understanding, all experiments were carefully designed and executed. The conclusions are in line with the presented data and discussion.

Still, the paper could potentially benefit from considering the following remarks.

1. In the introduction, it would be useful to clearly define what is considered as “the surface”, or the “surface/plasma interface”. While the Au film is only 80 nm thick,

surely this entire 80 nm cannot be considered as “the surface”. I imagine that a (time-dependent) temperature gradient must exist across the film.

We appreciate the suggestion and have revised our description of ‘surface/plasma interface’ throughout the text. Additionally, we have explicitly defined our measurement of the ‘surface temperature’ within the introduction; the surface here is defined as the temperature within the optical skin depth of the Au film (e.g., ~15 nm depth, at most).

2. Moreover, Au is used because of its noble character. However, at the nanoscale, Au is reactive, and in fact used as a catalyst. This reactivity is larger as the surface density of defects such as steps and kinks increase. Atomically flat surfaces are less prone to reaction. Thus, it would be useful to have a metric on the surface flatness.

We note that the Au film in this work is not atomically flat and was deposited via electron-beam evaporation, and is thus also polycrystalline. We have measured the surface roughness using a contact profilometer and it is ~3 nm RMS. This feature is noted in the Introduction of our revised text.

3. Very important for understanding the plasma/interface interaction is the knowledge of which species and densities thereof actually reach the surface. There is however no so much information provided on this; is the air (molecular species) excited or ionized? Is there any information on the electron densities and the electron energy distribution near the surface? Same question for the IEDF. I understand that the streamer reaches the Au film, but to me it is not so clear if only electrons and He⁺ ions (and other He-based excited species) reach the surface, or also other species.

In short, the air will mix with the He flow and lead to a wide range of excited and ionized species as well as reactive radicals. All of these species will impact the surface. We do not know the plasma density or electron temperature. However, based on our previous work, these values are estimated to be in the range of 10^{11} - 10^{12} cm⁻³ and approximately 2 eV, respectively. We have added a few sentences in the main article and in the supplemental material to address this aspect.

4. As a follow-up on the previous question, molecular plasmas behave very differently because a significant amount of the plasma power is used to excite species into vibrational and rotational levels. As most plasmas for applications use molecular gases, I wonder if the conclusions reached here would still be maintained in such case?

This is a good question. I think the answer here is an underwhelming, “perhaps”. Certainly, photon-driven desorption can happen in molecular gases. We expect changes

in the emission spectrum, including perhaps more emission in the VUV. As such the power balance at the surface could vary and change the heating/cooling balance. Similarly, the delivery of significantly more energetic species should change the power balance. That said, the internal energy possessed by rotationally/vibrationally excited species is typically much less than excited noble species. As a final point, the uptake of adsorbates could well change as radicals (e.g. O) will often adsorb more readily than their parent molecules (e.g. O₂).

5. In the beginning of the Discussion, the authors state “Although the notion of cooling resulting from an incident energy flux is seemingly counter-intuitive, ...” However, I would like to remark that even for a system in thermal equilibrium, microscopic balance and the 2nd law of thermodynamics dictate that the flux of species desorbing from the surface have a non-Maxwellian energy distribution, with a mean energy below that of the gas phase species, in case the impinging species have an energy-dependent sticking coefficient below 1 – aka, translational cooling in desorption. Of course, this does not mean that the surface is being cooled. I therefore wonder if the wording is or could be confusing to readers.

We appreciate the suggestion and agree that this statement could be confusing to potential readers; we have thus removed this statement.

REVIEWERS' COMMENTS

Reviewer #2 (Remarks to the Author):

This paper titled "Plasma Cooling" describes the cooling phenomenon of the gold surface when the He plasma jet impinges on it. The referee agrees that a novelty for this interesting phenomenon. As the authors described in the manuscript, this cooling phenomenon is very interesting and has potential for future applications. The authors answered the referees' comments properly, and the manuscript was improved to include a detailed explanation and discussions about their experiment. The revised manuscript can widely give impacts not only on process engineering but also on fundamental plasma physics. Thus, the referee recommends this work be published in Nature Communications.

Reviewer #3 (Remarks to the Author):

I have reviewed the comments of the referees – which are indeed all valid – and the comments of the authors to these comments.

As a theoretician, I must admit that I cannot judge or evaluate all of the implications of the measurements and how these measurements were carried out, but to the best of my understanding, the authors provided very reasonable responses to all issues raised. In particular with respect to my comments, I believe the authors answered satisfactorily.

I also believe this paper is of sufficient interest and novelty to deserve publication in Nature Communications.

REVIEWERS' COMMENTS

Reviewer #2 (Remarks to the Author):

This paper titled “Plasma Cooling“ describes the cooling phenomenon of the gold surface when the He plasma jet impinges on it. The referee agrees that a novelty for this interesting phenomenon. As the authors described in the manuscript, this cooling phenomenon is very interesting and has potential for future applications. The authors answered the referees' comments properly, and the manuscript was improved to include a detailed explanation and discussions about their experiment. The revised manuscript can widely give impacts not only on process engineering but also on fundamental plasma physics. Thus, the referee recommends this work be published in Nature Communications.

We greatly appreciate the reviewer's comments and suggestions through this review process.

Reviewer #3 (Remarks to the Author):

I have reviewed the comments of the referees – which are indeed all valid – and the comments of the authors to these comments.

As a theoretician, I must admit that I cannot judge or evaluate all of the implications of the measurements and how these measurements were carried out, but to the best of my understanding, the authors provided very reasonable responses to all issues raised. In particular with respect to my comments, I believe the authors answered satisfactorily.

I also believe this paper is of sufficient interest and novelty to deserve publication in Nature Communications.

We greatly appreciate the reviewer's comments and suggestions through this review process.